# Subcycle surface electron emission driven by strong-field terahertz waveforms

Shaoxian Li [1,2], Ashutosh Sharma[3], Zsuzsanna Márton[3,4], Priyo S. Nugraha[1,5], Csaba Lombosi[1], Zoltán Ollmann[1,4], István Márton [6,7], Péter Dombi[3,6], János Hebling[1,4,5] & József A. Fülöp[1,3] ✉

The advent of intense terahertz (THz) sources opened a new era when the demonstration of the acceleration and manipulation of free electrons by THz pulses became within reach. THz-field-driven electron emission was predicted to be confined to a single burst due to the single-cycle waveform. Here we demonstrate the confinement of single-cycle THz-waveform-driven electron emission to one of the two half cycles from a solid surface emitter. Either the leading or the trailing half cycle was active, controlled by reversing the field polarity. THz-driven single-burst surface electron emission sources, which do not rely on field-enhancement structures, will impact the development of THz-powered electron acceleration and manipulation devices, all-THz compact electron sources, THz waveguides and telecommunication, THz-field-based measurement techniques and solid-state devices.

The emission of electrons from atoms, molecules, and structured and bulk materials driven by strong oscillating electromagnetic fields has been the subject of extensive studies[1–6]. In case of photon energies below the ionization energy and moderately high-field strengths, multiphoton absorption or above-threshold ionization occurs. At very high-field strengths, the confining potential is significantly altered by the external field, which can lead to electron emission by tunneling through or liberation above the potential barrier.

Tunneling occurs over a time scale $T_t = \sqrt{2mW}/eE_0 = \gamma/\omega$ ($m$ is the electron mass, $e$ is the elementary charge, $W$ is the ionization energy or work function, and $E_0$ is the electric-field amplitude)[1–3]. The Keldysh (or adiabaticity) parameter $\gamma = 2\pi(T_t/T_{osc}) = \sqrt{W/2U_p}$ is often used as a scale parameter to characterize the different interaction regimes. Here, $T_{osc} = 2\pi/\omega$ is the duration of one oscillation cycle of the driving field and $U_p = e^2E_0^2/4m\omega^2$ is the electron ponderomotive energy (average kinetic energy of a free electron in the oscillating laser field). In the strong-field (also called adiabatic or quasistatic) regime $\gamma < 1$, the tunneling time becomes shorter than half of the field oscillation period, and the maximum energy transfer

to an electron in the absence of rescattering ($2U_p$) exceeds the ionization energy.

The transition regime between multiphoton absorption and tunneling emission from metal surfaces was studied with picosecond laser pulses in the intensity range of 5 GW/cm² to 120 GW/cm² (ref. 4,5). Strong-field plasmonic photoemission was observed, driven by femtosecond laser pulses and utilizing field enhancement in metallic nanoparticles[7], nanotips[8], and films[9,10]. In the latter case, the favorable scaling of the electron ponderomotive energy with the smaller mid-infrared frequency ($U_p \propto \omega^{-2}$) enabled electron emission at intensities below 1 GW/cm² (ref. 9).

A particularly interesting regime is ionization by very short, few-or single-cycle laser pulses, where the electron emission can sensitively depend on the carrier-envelope phase (CEP)[11,12]. Photoelectron spectra from above-threshold ionization were shown to exhibit CEP dependence[13]. The generation of isolated attosecond pulses by optical field ionization of noble gas atoms and high-harmonic generation relies on waveform-controlled few-cycle driver laser pulses[14]. CEP dependence was predicted for ponderomotive surface-plasmon

[1]Szentágothai Research Centre, University of Pécs, 7624 Pécs, Hungary. [2]Center for Terahertz Waves and College of Precision Instrument and Optoelectronics Engineering, and the Key Laboratory of Optoelectronics Information and Technology, Ministry of Education, Tianjin University, 300072 Tianjin, China. [3]ELI-ALPS Research Institute, ELI-HU Non-Profit Ltd., 6728 Szeged, Hungary. [4]Institute of Physics, University of Pécs, 7624 Pécs, Hungary. [5]HUN-REN-PTE High-Field Terahertz Research Group, 7624 Pécs, Hungary. [6]Wigner Research Centre for Physics, 1121 Budapest, Hungary. [7]Institute for Nuclear Research (Atomki), 4001 Debrecen, Hungary. ✉e-mail: jozsef.fulop@eli-alps.hu

electron acceleration[15], and experimentally demonstrated utilizing field enhancement at gold nanotips[16] and nanoparticles[3].

Recently, the study of electron emission phenomena has been extended to the terahertz (THz) regime[17], where intense single-cycle pulses have proven unprecedented discovery potential in a broad range of science[18,19]. The transition regime, and even the strong-field regime, can be easily accessed with moderately high, µJ-level THz pulse energies owing to low frequencies and single-cycle waveforms. In strong-field ionization of excited atoms, the suppression of adiabatic over-the-barrier ionization was found due to the extended times required for highly excited electrons to leave the binding potential[20]. THz-field acceleration up to 1.5 keV and control of propagation direction was proposed for electrons generated by femtosecond laser-driven surface plasmons[21]. THz-induced transient increase of the optical reflectivity of a thin gold film was observed and explained by the influence of electron tunnel emission[22]. THz-driven electron field emission was reported up to $\gamma \approx 0.01$ across a split-gap dipole antenna[23] and from metallic microantenna[24], where THz-driven electron acceleration, nitrogen plasma generation, and UV emission were described[24]. Electron energies exceeding 5 keV[25] and bunch charges of $10^6$ electrons per pulse[26] were observed from THz-driven metallic nanotips[27]. Electron emission driven by strong-field single-cycle THz waveforms was predicted to be confined to a single burst[25], however, a direct experimental demonstration is still lacking.

Here we report on subcycle surface electron emission driven by strong-field THz waveforms. The THz-pulse-induced electron emission from a BeO surface is experimentally studied. The utilization of intense single-cycle THz pulses enabled easy access to the transition regime, without the need for field-enhancement structures. Electron emission was observed at a threshold field strength as low as 40 kV/cm. This is in contrast to previous works[23,24], where very large field-enhancement factors of 170× and 32× were needed to achieve detectable emission at extremely high fields of about 10 MV/cm and 2 MV/cm, respectively. In

the present work, signatures of subcycle electron emission dynamics were studied through varying the field polarity direction. Effects of the emitter's surface roughness on the electron emission characteristics are also discussed.

## Results

### THz-waveform-driven surface electron source

The experimental setup (Fig. 1a) for the THz-driven surface electron source consisted of an intense single-cycle THz pulse source with THz beam collimation and focusing (see Methods: THz pulse source), and the experimental chamber (see Methods: Interaction chamber), containing the THz-driven BeO cathode surface and an integrated electron multiplier chain for the highly sensitive detection of the emitted electrons. BeO is frequently used in electron-emitting devices, such as electron multipliers, and may be of interest for future THz-field-driven electron source technology. The experimental chamber was rotatable about the THz beam propagation axis to enable electron emission measurements at continuously variable polarization angles $\varphi$ of the THz electric field (Fig. 1b). This included flipping the polarity of a p-polarized THz pulse from $\varphi = 0°$ to $\varphi = 180°$ and vice versa. We opted for chamber rotation instead of using a rotatable waveplate, to avoid any distortion of the THz pulse waveform due to the dispersion of retardation.

The THz pulses had a single-cycle waveform with an electric-field amplitude $E_0$ at the BeO cathode (Fig. 1c). The field maxima were different in the leading and trailing half-cycles: It was $E_1 = E_0 \times 0.64$ in the leading, and $E_2 = E_0$ in the trailing half-cycle, respectively. This difference was important for distinguishing between the two half cycles in the electron emission process, which will be discussed below. The maximum applied value of the electric-field amplitude was 129 kV/cm (see Methods: THz beam and pulse characterization). With the THz mean frequency of 0.38 THz (inset in Fig. 1c) and an approximate work function of $W = 4.1$ eV for the BeO surface[28], the corresponding minimum applied value of the Keldysh parameter was $\gamma = 1.26$.

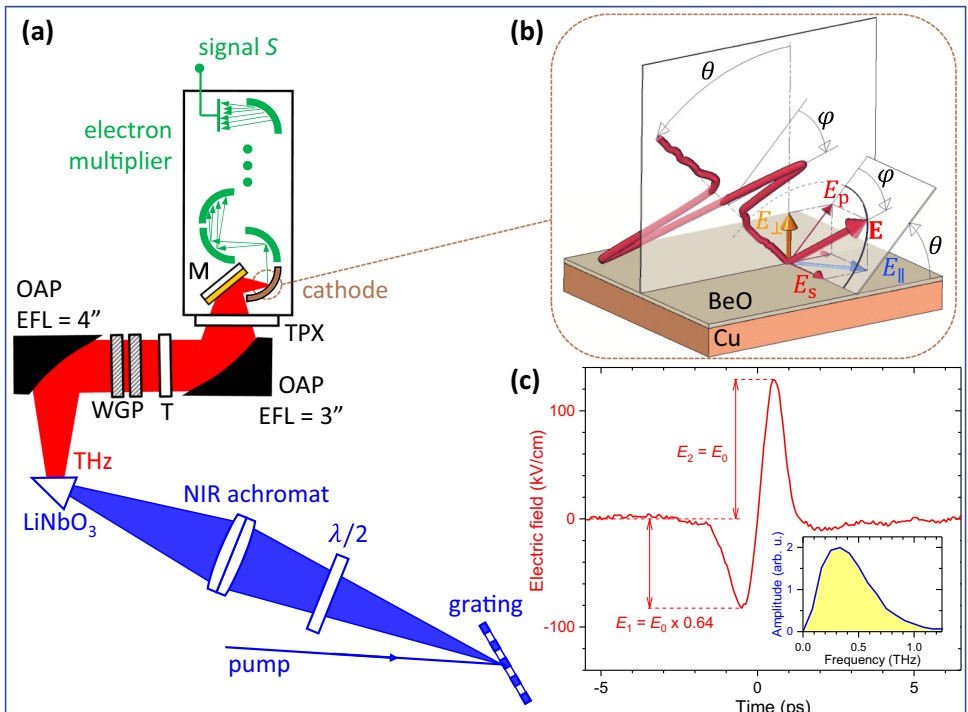

**Fig. 1 | The THz-waveform-driven surface electron source. a** Layout of the experimental setup. $\lambda/2$: half-wave retardation plate, NIR: near-infrared, OAP: off-axis parabolic mirror, EFL: effective focal length, WGP: wiregrid polarizer, T: Teflon plate, TPX: polymethylpentene window, M: mirror. **b** Interaction geometry at the BeO cathode surface. $E_\perp, E_\parallel, E_s, E_p$: components of the THz electric field **E** at the cathode surface, $\varphi$: polarization angle, $\theta \approx 45°$: angle of incidence of the THz beam. **c** Waveform of the THz pulse with electric-field amplitude $E_0$. The maximum field in the leading (trailing) half-cycle is $E_1 = E_0 \times 0.64$ ($E_2 = E_0$). The electric-field strength scale corresponds to the highest field used in the experiment. The inset shows the calculated amplitude spectrum. For details see Methods: THz beam and pulse characterization.

For field-driven processes in the single-cycle regime, it is important to know the polarity of the THz pulse in the laboratory frame. In the present case of electron emission from a cathode surface, this requires the measurement of the direction of the electric-field vector with respect to the cathode surface within each half-cycle of the oscillating field. The THz polarity was calibrated by comparison with a static (DC) field (Fig. 2). Details are given in Methods: Calibration of the THz field polarity.

It is the local near field at the cathode surface, which directly governs the electron emission process. The nanometer-scale surface topography affects the near field and can lead to strong variations in it. Characterization of the cathode surface topography by atomic force microscopy (see Supplementary Information: Characterization of the BeO surface) and subsequent numerical simulations based on the finite-element method showed typical local maxima of the field-enhancement factor of up to about 2 to 4.

## THz-field polarity-controlled electron emission

The THz-driven electron emission signal from the BeO cathode surface was recorded as a function of the THz electric-field amplitude (peak electric-field strength) $E_0$. The amplitude was varied from 40 kV/cm to 129 kV/cm. The THz field was p-polarized, i.e., it was in the plane of incidence. Figure 3a shows the measured electron emission signal ($S$)

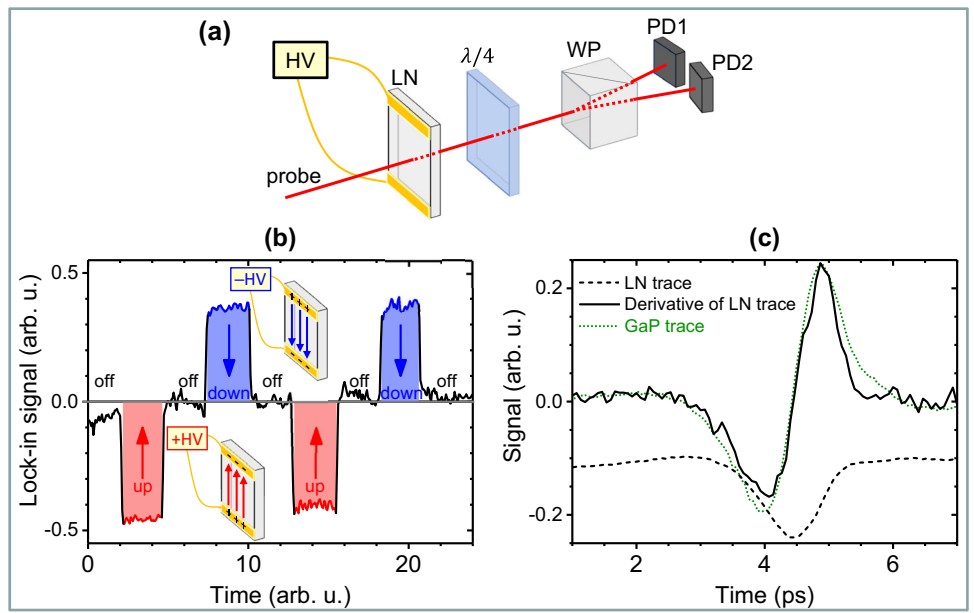

**Fig. 2 | Calibration of the THz field polarity. a** Schematic drawing of THz polarity calibration by comparison with a static (DC) field, utilizing a LiNbO₃ (LN) crystal for electro-optic sampling (EOS). PD1, PD2: photodiodes, HV: high DC voltage. **b** Differential signal of the two photodetectors PD1 and PD2 when the applied HV was off, on (up), and reversed (down). The red arrows show the DC electric-field direction in the laboratory frame. **c** EOS trace of the THz pulse measured with the LN crystal (black dashed line) and its time derivative (black solid line). The EOS trace obtained with the GaP crystal (green dotted line) is shown only for waveform comparison and was not used for the polarity calibration.

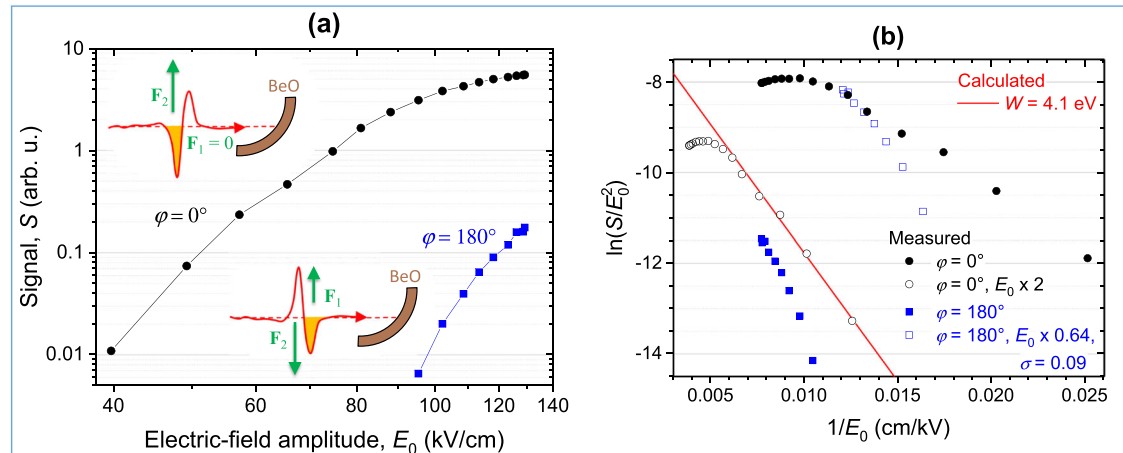

**Fig. 3 | THz-field polarity-controlled electron emission. a** Measured electron emission signals as functions of the THz electric-field amplitude $E_0$ for p-polarized cases with the opposite polarities φ = 0° (full black circles) and φ = 180° (full blue squares). The two insets show the BeO cathode irradiated by the THz waveforms of opposite polarities (red solid curves), the active half-cycle (orange filling), and the effective force acting on the electrons (green arrows). **b** Fowler-Nordheim plot of the measured data (full symbols, same data as in part **a**). The empty blue squares show the measured data for φ = 180° with the electric-field strength scaled by the half-cycle asymmetry factor of 0.64 and an electron backscattering coefficient of σ = 0.09. The empty black circles show the measured data for φ = 0° with the electric-field strength scaled by the effective field-enhancement factor of 2. The red solid line shows the calculated emission current with the BeO work function of 4.1 eV. More details are given in the text.

as a function of the THz peak electric-field strength in a log-log plot. In the first case (black circle symbols in Fig. 3a), the force exerted by the electric field on the electrons pointed into the BeO cathode surface during the leading half-cycle of the THz pulse and the electric field pointed outward (see the red curve in the upper-left inset in Fig. 3a). During the trailing half-cycle, the force (field) reversed to point outward (inward). The green arrow in the inset indicates the force during the trailing half-cycle. This case is labeled by a polarization angle of $\varphi = 0°$. The polarization angle is defined here by the angle of the electric-field vector in the leading half-cycle of the THz pulse, as measured from the incidence-side (vacuum-side) half of the plane of incidence (Fig. 1b). It can be seen from Fig. 3a that a detectable electron emission signal occurs already at a moderate THz field strength of 40 kV/cm, corresponding to $\gamma \approx 4.1$. The electron emission signal increases in a nonlinear way with increasing THz electric-field strength. We note that fitting conventional multiphoton photoemission forms $J \propto |E_0|^{2n}$ to the measured data gives logarithmic slopes of $n = 4.2$ for the low-field (below 60 kV/cm) and $n = 0.62$ for the high-field (above 110 kV/cm).

In the second case (blue square symbols in Fig. 3a, labeled by $\varphi = 180°$), the THz field polarity was reversed so that the force (field) would point outward (inward) the BeO cathode surface during the leading half-cycle, and opposite during the trailing half-cycle (lower inset in Fig. 3a). Remarkably, the electron emission signal was reduced by about two orders of magnitude, as compared to the $\varphi = 0°$ case. A clearly measurable current signal was seen only above 95 kV/cm electric-field amplitude, which corresponds to $\gamma \approx 1.7$. In this case, the logarithmic slope obtained from the multiphoton photoemission fit resulted in $n$ values varying from 8.2 to 3.7 for the lower- and the higher-field part of the measured data, respectively.

## Discussion

For a qualitative interpretation of the observations, the Fowler-Nordheim theory of cold-field electron emission[29] was used to estimate the electron yield from the BeO cathode surface (see Supplementary Information: Theoretical model of the field-driven electron emission). In a single-cycle pulse, electrons are emitted only during one of the two half-cycles, the active half-cycle (orange filling in the insets of Fig. 3a), during which the field points inward the surface and the force on the electrons outward. The approximately two-orders-of-magnitude difference between the electron emission signal at $\varphi = 0°$ and $\varphi = 180°$ can be partially explained by the different maxima of the electric field in the leading and trailing half cycles of the THz pulse. According to the waveform measurement (Fig. 1c), the maximum field strength at the BeO surface in the leading half-cycle, $E_1$, was about 64% of that in the trailing half-cycle, $E_2$, i.e., $E_1 = E_2 \times 0.64$. At $\varphi = 0°$ polarity, the stronger trailing half-cycle was active, resulting in a larger electron emission signal. In contrast, at $\varphi = 180°$ polarity, the weaker leading half-cycle was active and the electron emission signal was much smaller (at the same values of the pulse electric-field amplitude $E_0$).

In order to gain more insight into the electron emission process, the measured electron emission signal ($S$) is plotted in a Fowler-Nordheim plot in Fig. 3b (full symbols). This is a linearized representation, where the quantity $\ln(S/E_0^2)$ is plotted as function of $E_0^{-1}$. Within the linear response range of the electron multiplier, the measured signal $S$ is proportional to the surface- and time-integrated emission current density.

The electron yields, measured at the two opposite field polarities, can be more directly compared by using the electric-field maximum of the respective active half-cycle, rather than the pulse amplitude $E_0$ for both. As mentioned above, the electric-field maxima are about $E_1 = E_0 \times 0.64$ and $E_2 = E_0$ for the leading and trailing half cycles, respectively (Fig. 1c). When scaling the field strength by the half-cycle asymmetry factor of 0.64 for the $\varphi = 180°$ polarity, this dataset moves closer to the data for $\varphi = 0°$ polarity. However, a significant

discrepancy remains between them (see Supplementary Figure 5 in Supplementary Information: Theoretical model of the field-driven electron emission). A better agreement can be achieved when electron backscattering is taken into consideration. In case of $\varphi = 180°$, most of the electrons are emitted near the crest of the leading (weaker) half-cycle. The emitted electrons follow returning trajectories and are driven back to the BeO surface by the reversed force during the trailing half-cycle ($F_2$, bottom inset in Fig. 3a). The returning electrons either recombine into the surface, or they are backscattered with a probability given by the backscattering coefficient $\sigma$. This reduces the electron emission signal by the factor $\sigma$. When both the half-cycle asymmetry factor of 0.64 and a backscattering factor of about $\sigma = 0.09$ are taken into account for $\varphi = 180°$, a reasonable agreement can be achieved with the $\varphi = 0°$ dataset (cf. empty square and full circle symbols in Fig. 3b). More details on backscattering, including numerical calculation based on the time-dependent Schrödinger equation, are given in Supplementary Information: Theoretical model of the field-driven electron emission.

For analyzing the dependence of the emission signal on the electric-field strength, the $\varphi = 0°$ polarity is considered here, because it is not affected by electron backscattering. The calculated dependence of the electron emission current on the electric-field maximum is shown by the red solid line in Fig. 3b for $\varphi = 0°$. In the calculation, a value of 4.1 eV was used for the work function of the BeO cathode surface[28]. A significant discrepancy between the calculation and the measurement (full black circles in Fig. 3b) was found. However, by rescaling the field strength from $E_0$ to $E_0 \times 2$, the measured data for $\varphi = 0°$ come to a reasonably good agreement with the calculation (cf. the empty circles and the solid red line in Fig. 3b). This behavior hints at a possible small average field-enhancement effect with a surface-averaged effective field-enhancement factor of $\langle\eta\rangle \approx 2$. Because of this and the proportionality $\gamma \propto E_0^{-1}$, the effective values of the Keldysh parameter were about twice as small than estimated from the field strength measurements. Thus, the minimum effective value available in these experiments was $\gamma_{eff} = \gamma/\langle\eta\rangle \approx 0.63$, in contrast to the enhancement-free minimum of $\gamma \approx 1.26$ for the incident field (see the subsection THz-waveform-driven surface electron source). The origin of this enhancement effect is the roughness of the BeO cathode surface, which leads to a locally varying enhancement factor, as discussed below.

Note that for a field dependence of the current, obeying the Fowler-Nordheim model, the slope in the Fowler-Nordheim plot is proportional to $W^{3/2}/\langle E\rangle$. The effective field for $\varphi = 0°$ polarity is $\langle E\rangle = E_0 \times \langle\eta\rangle$. The effect of field enhancement can also be regarded as a modification of the effective work function of the surface from $W = 4.1$ eV to $W \times \langle\eta\rangle^{-2/3} \approx 2.5$ eV.

To provide an independent characterization of the field enhancement, topography measurements of the BeO cathode surface were carried out, using atomic force microscopy (AFM). Figure 4a shows the AFM topography data, with 150 nm resolution, for an example area of $30 \times 30 \mu m$. Granular structures can be seen, with typical grain sizes on the μm-nm length scales and height variations of several tens to hundreds of nanometers. Subsequently, based on the topography data, numerical simulations of the THz near field were performed to quantify the resulting field enhancement and its local variations (see Methods: Calculation of the THz near field). Figure 4b shows the calculated local field-enhancement factor $\eta(x,y)$ over a selected $7.5 \times 7.5 \mu m$ area, as indicated by the dashed square in Fig. 4a. In this area, $\eta$ varies from 0.3 to 2.4 at a height of 50 nm above the surface. Figure 4c shows the calculated enhancement $\eta(x,z)$ in a plane vertical to the BeO surface, as indicated by the horizontal dotted line in Fig. 4b. At locations of stronger positive curvature, typically at grains, maxima for the field-enhancement factor up to 2 to 4 were found at the surface. These values are in good agreement with the estimation given above for the effective field enhancement (about 2), which was solely

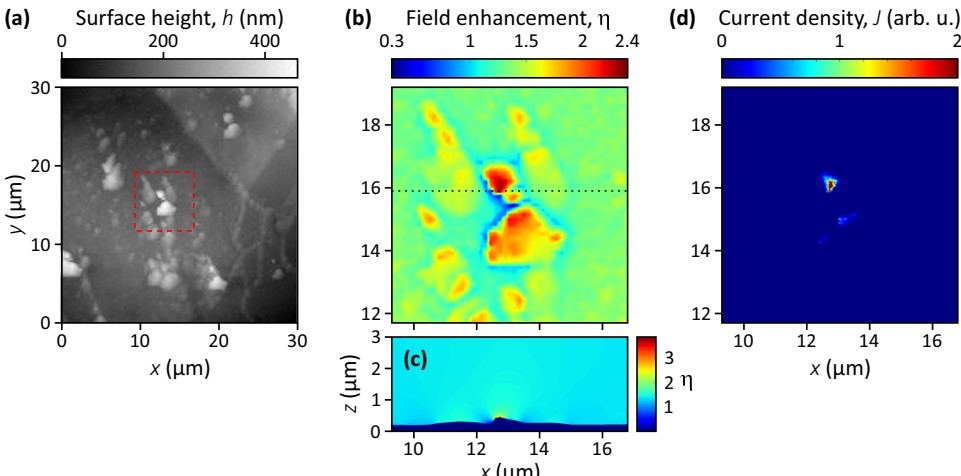

**Fig. 4 | Field enhancement and local electron emission characteristics of the THz-waveform-driven surface electron source. a** Surface topography of the BeO cathode, measured by AFM. The full area shown is $30 \times 30\,\mu m$ in size and has a granular surface structure. **b** Local variation of the field enhancement, $\eta(x,y)$, calculated by the finite-element method at a height of 50 nm above the surface, over the $7.5 \times 7.5\,\mu m$ sized area indicated by the dashed square in part **a**). **c** The

calculated field enhancement, $\eta(x,z)$, in a plane perpendicular to the mean cathode surface, as indicated by the dotted horizontal line in part **b**. **d** The calculated surface distribution of the current density $J(x,y)$ over the $7.5 \times 7.5\,\mu m$ sized area indicated by the dashed square in part **a**. A polarity angle of $\varphi = 0°$ (trailing half-cycle active) and an electric-field amplitude of $E_2 = E_0 = 105\,kV/cm$ were assumed.

based on fitting the experimental data with a Fowler-Nordheim model (Fig. 3b).

The calculated local electron current density $J(x,y)$ is shown in Fig. 4d for the same selected $7.5 \times 7.5\,\mu m$ surface area. It was calculated with the Fowler-Nordheim formula (see Supplementary Information: Theoretical model of the field-driven electron emission), from the calculated local electric-field data at the cathode surface (see Methods: Calculation of the THz near field). A strong localization of electron emission is clearly visible in the surface distribution of the current. Such emitting hot spots cover a relatively small fraction of the total cathode area and are located around the local maxima of the field enhancement because electron emission is a highly nonlinear and therefore highly selective process. The effective field enhancement (about 2) is somewhat smaller than the highest local maxima (2 to 4), because the former is an average value over that part of the surface where significant electron emission takes place. Thus, the AFM measurements and the numerical calculation of the near field clearly confirmed that the cathode surface roughness could yield the required field enhancement. The calculation of the current density gave additional insight into the surface distribution of the emission.

In summary, we investigated electron emission from a BeO surface induced by intense single-cycle THz pulses in the transition regime. THz pulses with 0.38 THz mean frequency were focused to peak field strengths up to 129 kV/cm at the BeO surface, corresponding to a Keldysh parameter of $\gamma \approx 1.26$. A threshold field of about 40 kV/cm ($\gamma \approx 4.1$) was observed to induce electron emission. This is in contrast to previous works (see for example refs. 23 and 24) based on very strong-field enhancements of 170× and 32× to achieve detectable emission at extremely high local fields of about 10 MV/cm and 2 MV/cm, respectively. The reason is the high detection sensitivity of the electron multiplier used here, the large total number of electrons due to the large emitting surface, and a small (2×) effective field enhancement due to surface roughness. The latter led to an effective minimum value of the Keldysh parameter of $\gamma_{eff} \approx \gamma/2 \approx 0.63$. The electron yield was two orders of magnitude larger when the electron emission was confined to the stronger half-cycle of the pulse, as compared to the polarity-flipped case with emission confined to the weaker half-cycle. In the latter case, a threshold field of 95 kV/cm ($\gamma \approx 1.7$) was found, which is more than two times higher than for the opposite polarity. These findings can be of importance for a broad

range of emerging technologies. The applications include THz-driven electron accelerators and manipulation devices[30], THz-driven electron sources, THz waveguides, and telecommunication, and THz-field-based measurement technologies.

## Methods
### THz pulse source
For the generation of intense single-cycle THz pulses, the tilted-pulse-front technique was used[31,32]. Pump pulses of 200 fs duration, 1.03 μm wavelength, 1 kHz repetition rate, and pulse energy up to 4 mJ were diffracted from a reflection grating and then imaged by a near-infrared achromatic lens of 20 cm focal length into a $LiNbO_3$ prism. The emitted vertically polarized THz beam was collimated by an off-axis parabolic mirror (OAP) of 4" effective focal length (EFL), and focused by another OAP of 3" EFL. Two wiregrid polarizers were inserted into the collimated THz beam for the convenient control of the THz electric-field amplitude. A thick Teflon plate blocked the scattered pump light and its second harmonic.

### Interaction chamber
The focused THz beam entered the experimental chamber through a polymethylpentene (TPX) window and was reflected by a gold mirror of 1 cm × 1 cm in size. The chamber was evacuated to a pressure of about $4 \times 10^{-6}$ mbar. The angle of incidence of the THz beam at the mirror surface was about 45°. The gold mirror had a dielectric protective coating of about 100 nm thickness which inhibited possible THz-field-driven electron emission from gold at the field strengths used here. The mirror directed the THz beam onto the THz-driven BeO cathode at approximately 45° angle of incidence. The emitted electron pulses were amplified by a modified electron multiplier (Hamamatsu R595), operated at a bias voltage of 2.5 kV. In the modified electron multiplier, the first dynode was replaced with the gold mirror mentioned above, the second dynode acted as the THz-field-driven cathode, and the subsequent dynodes were biased through a voltage divider circuit and used for electron multiplication. The integration of the THz-driven cathode and the electron multiplier resulted in a very compact experimental configuration and enabled the highly efficient detection of the emitted electrons. The material of the cathode was BeO, which was coated onto a Cu substrate (see Supplementary Information: Characterization of the BeO surface). The output current

pulse was converted to a voltage pulse through a 50 Ω impedance and monitored with an oscilloscope. The time integral of the voltage pulse was recorded.

The BeO cathode surface had a cylindrical shape with a curvature radius of about 1 cm. As the THz beam, focused onto the cathode, had a relatively small waist radius of about 1.2 mm and a depth of focus of several centimeters due to astigmatism (see Methods: THz beam and pulse characterization), the cathode surface curvature had only a slight effect on the measured electron emission characteristics. The roughness of the BeO surface had a significant effect on electron emission. The characterization of the surface is described in more detail in Supplementary Information: Characterization of the BeO surface.

### THz beam and pulse characterization

The THz pulse energy was measured by a calibrated pyroelectric detector (Sensor- und Lasertechnik, THZ 20). A maximum energy of 2.34 µJ was used at the BeO surface.

THz beam profiles were recorded with a pyroelectric camera (Ophir, Pyrocam III) at different positions along the propagation axis. The beam waist radius in both the horizontal and the vertical directions was about 1.2 mm (corresponding to $1/e^2$ values of the peak intensity). Due to astigmatism, the distance between the horizontal and vertical foci was about 2 cm. This resulted in an extended focal region and a reduced variation of the peak field strength across the BeO cathode surface.

THz waveforms were recorded by electro-optic sampling (EOS) in a 1-mm thick (110)-cut GaP crystal by using a small portion of the laser energy for the sampling pulses and balanced photodetection with a lock-in amplifier. The measured waveform is shown in Fig. 1c for the beam propagation coordinate corresponding to the BeO cathode position. The field strength and the single-cycle, nearly sine-like waveform changed only slightly over a propagation length of about 2 cm. This ensured a reasonably good uniformity of the THz waveform across the illuminated spot at the BeO cathode surface for all THz polarization directions used in the experiment. The electric-field amplitude was obtained by combining the measured THz waveform, beam size, and pulse energy.

### Calibration of the THz field polarity

The THz polarity was calibrated by comparison with a static (DC) field, utilizing an insulator LiNbO$_3$ (LN) crystal for EOS (Fig. 2a). First, in the absence of the THz field, a high DC voltage (HV) of 4.4 kV and controlled polarity was applied to the LN crystal through the two electrodes near the top and bottom edges of the crystal. The optical sampling pulse was sent through the crystal. When the HV was off, the two photodetectors were balanced and the output from the lock-in amplifier was zero. When an HV was applied with an upward-pointing electric field, the output from the lock-in amplifier was negative. When the HV was reversed, the output changed sign. Figure 2b shows the lock-in output of two such cycles with the HV manually switched on, off, and reversed. This output directly reflects the phase change of the probe passing through the LN crystal.

During the next measurement steps, the HV was switched off and the THz pulse passed through the LN detector crystal. The optical sampling pulse propagates much faster through the crystal than the THz pulse and sweeps over the portion of the THz pulse that lies inside the crystal[33,34] (see Supplementary Information: Calibration of the THz field polarity). The THz electric-field waveform, directly comparable in polarity to the DC field, was obtained by taking the time derivative of the EOS trace originating from the input region of the crystal. Because the THz signal changed from negative to positive (Fig. 2c), a comparison with the HV bias (Fig. 2b) indicated that in the laboratory frame, the THz electric-field pointed upwards (downwards) in the leading (trailing) oscillation half-cycle. Based on this result and the geometry of the experimental setup, and taking into account the phase shift

upon reflection at the gold mirror, the THz electric-field polarity could easily be traced to the cathode surface. More details on field calibration are given in Supplementary Information: Calibration of the THz field polarity.

### Calculation of the THz near field

The THz near field at the cathode surface was calculated by the finite-element method. The measured surface topography data, obtained by atomic force microscopy (see Supplementary Information: Characterization of the BeO surface), were used to define the boundary conditions. For the simulations, two types of software were used: the FDTD 3D Electromagnetic Simulator (Lumerical, Inc.) for time-domain calculations and the COMSOL Multiphysics® (COMSOL AB) software for frequency-domain calculations. The latter was carried out at 0.3 THz and 0.5 THz frequencies. Both methods gave very similar field-enhancement values. The field enhancement was defined as the ratio of the THz electric-field amplitude at the surface and the amplitude of the incoming field. Calculation results are shown in Fig. 4b, c.

## Data availability

Data underlying the results presented in this paper are available from the corresponding author upon request. The experimental data generated in this study have been deposited in the Figshare database under the accession code https://doi.org/10.6084/m9.figshare.24065715[35].

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

## Acknowledgements
The authors acknowledge the seminal contribution of the late Győző Farkas (Wigner Research Centre for Physics, Budapest) to this research. He raised the question of strong-field electron emission from surfaces with the help of intense THz pulses. National Research, Development and Innovation Office (ANN 139483, KKP 137373, 2018-1.2.1-NKP-2018-00012, TKP2021-NVA-04, TKP2021-EGA-17). European Union (GINOP-2.3.6-15-2015-00001, FET Open grant "PETACom"). European Regional Development Fund (GINOP-2.3.6-15-2015-00001).

## Author contributions
J.H. proposed the idea of the experiment. J.H. recognized the importance of the polarity order of the THz pulses. J.H. and J.A.F. conceived the method to determine the polarity of THz pulses. J.H., P.D. and J.A.F. conceived the experimental setup. S.L., P.S.N., C.L., Z.O., I.M., P.D. and J.A.F. contributed to constructing the experimental setup. S.L., P.S.N., C.L., Z.O., I.M. and J.A.F. performed the measurements. Surface characterization was carried out by Z.M. The data were analyzed by S.L., A.S. and J.A.F. The paper was written by S.L., A.S. and J.A.F. and reviewed by all authors.

## Funding

## Competing interests
The authors declare no competing interests.
