## [Peer Review File · Nature Communications]

Subcycle surface electron emission driven by strong-field terahertz waveformsReviewers' Comments:

Reviewer #1:

The manuscript entitled “Subcycle surface electron emission driven by strong-field terahertz waveforms...” by S. Li reported the surface electron emission from Cu-BeO excited by single burst of THz Waveforms. The authors demonstrated the electron emission mechanism is multiphoton photoeffect; For the case of field dependence for opposite polarities of excitation, the authors demonstrated that cold-field electron emission are excited only by one of the two half cycles, under a single-cycle pulse. In addition, the authors furtherly analyzed the surface electron emission dependent field polarisation angle, from which they conclude that the double maxima amplitude of the electron emission dependent the electric field polarisation angle can be attributed to the effect of surface roughness. This paper presents an interesting electron emission from Cu-BeO excited by a single-cycle THz pulse, and a details analysis effect on field polarisation angle and surface roughness, which is significant to design of THz-driven electron devices. However, I still have a few concerns that needed to be addressed before publication, which are listed as follows:

1. Illustration for the experimental chamber geometry with the THz electric field polarisation angles φ indicated would be useful. It would also be useful to understanding the THz-driven electron emission from the Cu-BeO cathode surface during the leading or the trailing half cycle of the THz pulse.
2. The electron emission from Cu-BeO was observed at a threshold field strength as low as 40 kV/cm, which is different from previous detection work with high fields of about 10 MV/cm and 2 MV/cm. the reason can be the large total number of electrons due to the large emitting surface and a small average field enhancement due to surface roughness, which is insufficient. Such as in Figure 3b, the surface roughness doesn't affect the intensity of electron emission. The electron emission signals from reference substrate Cu and BeO are necessary.
3. The thickness of Cu on Cu-BeO is useful. It is important to determine the electron emission mechanism.
4. The experimental data (Figure 3a) is asymmetric about $\varphi=0^\circ$, which is attributed to asymmetric condition on surface Cu-BeO. The supplemental data about the dependence of electron emission on azimuthal angle of Cu-BeO should be given, which are important experimental evidence.

Reviewer #2:

Remarks to the Author:

In the manuscript "Subcycle surface electron emission by strong-field terahertz waveforms" by Shaoxian Li et al., the authors reveal the electron emission from a Cu-BeO surface driven by intense THz pulses. This process was observed at remarkably low field strengths of 40 kV/cm without the need for field enhancements generated by nanostructures, for example. A potential explanation for this low threshold lies in the large number of detected electrons owing to the large area that was illuminated. The single-cycle nature of the THz pulses and the significant change in electron yield upon reversal of the field polarity confirm that the emission of the electrons is confined to the halfcycle exerting a force onto the electrons that points away from the surface. Discrepancies between modelling and experiments suggest that the following halfcycle of the transient may drive some of the electrons back to the surface, thereby lowering the electron yield. The data quality is convincing and sufficient details are given for the reader to be able to reproduce the results, in principle. Even though the manuscript is well written and interesting, and the experiments were carefully carried out, I cannot support publication in Nature Communications for the following reasons:

1. The conclusions drawn from the experimental data remain very vague. To describe the dependence of the detected number of electrons as a function of the electric field, the authors need to make several assumptions. On the one hand, there is a discrepancy (0.43 instead of the expected scaling factor of 0.65) used for the relative field strength of the two half cycles. The authors suggest that this could originate from electrons being driven back to the surface. This hypothesis is, however, not quantified by simulations (based on the time-dependent Schrödinger equation, for example). On the other hand, all electric fields additionally need to be scaled by a factor of two to retrieve the correct work function in the Fowler-Nordheim plot. The authors suggest that the surface roughness could yield the required field enhancement

2. The field enhancement resulting from the surface roughness is not quantified reliably on the relevant length scales. The maxima in the polarization dependence on the number of emitted electrons suggest that a significant surface roughness is present. To describe their data, a characteristic parameter of $\sim 15^\circ$ is required. Yet, the authors only provide optical images on the micrometer-scale resulting in a significantly smaller value of 2.6° , for which only a single maximum is expected according to their theoretical model. For a high-ranking journal such as Nature Communications, topography measurements on the nanoscale (atomic force microscopy etc.) should be conducted for a reliable quantification of the roughness parameter. Based on these insights, numerical simulations (finite element method, for example) should subsequently be performed to quantify the resulting field enhancement in the THz spectral range. This would also resolve some of the points addressed in (1).

3. How accurately is the work function of Cu-BeO known? The authors only refer to a personal communication with the manufacturer Hamamatsu? Can the possibility of a significantly smaller work function (< 4.1 eV) reducing the required field enhancement be ruled out? Additional AFM measurements and numerical simulations (see my previous point) would also help clarify this point. Are there any measurements or literature values?

4. Some minor points:

- The electric field in Fig. 1b could be calibrated to kV/cm
- A justification why Cu-BeO was used is missing
- In the introduction (line 30), references should be added for the extensive studies that are mentioned

While the manuscript provides new and interesting results, the open questions and unknowns listed above are limiting the potential impact of this work. Unless the authors can significantly revise the manuscript and provide a more quantitative explanation of their experimental results, I cannot support publication in Nature Communications, unfortunately.

Response to Reviewer 1

We thank the Reviewer for the critical reading of the manuscript and for the valuable and constructive comments, which helped us to improve significantly the manuscript. Furthermore, we appreciate the Reviewer's opinion of finding our results interesting. In our point-by-point response below, the Reviewer points are typeset in blue and our responses in black. The description of the corresponding changes in the manuscript are typeset in red.

Reviewer point:

The manuscript entitled "Subcycle surface electron emission driven by strong-field terahertz waveforms..." by S. Li reported the surface electron emission from Cu-BeO excited by single burst of THz Waveforms. The authors demonstrated the electron emission mechanism is multiphoton photoeffect; For the case of field dependence for opposite polarities of excitation, the authors demonstrated that cold-field electron emission are excited only by one of the two half cycles, under a single-cycle pulse. In addition, the authors furtherly analyzed the surface electron emission dependent field polarisation angle, from which they conclude that the double maxima amplitude of the electron emission dependent the electric field polarisation angle can be attributed to the effect of surface roughness. This paper presents an interesting electron emission from Cu-BeO excited by a single-cycle THz pulse, and a details analysis effect on field polarisation angle and surface roughness, which is significant to design of THz-driven electron devices. However, I still have a few concerns that needed to be addressed before publication, which are listed as follows:

1. Illustration for the experimental chamber geometry with the THz electric field polarisation angles ϕ indicated would be useful. It would also be useful to understanding the THz-driven electron emission from the Cu-BeO cathode surface during the leading or the trailing half cycle of the THz pulse.

Author response:

The requested figure was prepared. It shows the THz electric field polarization angle and the components of the field. It was included into the revised manuscript as the new Fig. 1b. The caption of Fig. 1 was adapted to the new figure and it is called out in the main text.

Reviewer point:

2. The electron emission from Cu-BeO was observed at a threshold field strength as low as 40 kV/cm, which is different from previous detection work with high fields of about 10 MV/cm and 2 MV/cm. The reason can be the large total number of electrons due to the large emitting surface and a small average field enhancement due to surface roughness, which is insufficient. Such as in Figure 3b, the surface roughness doesn't affect the intensity of electron emission. The electron emission signals from reference substrate Cu and BeO are necessary.

Author response:

2.A

As pointed out by the Reviewer, the small effective field enhancement is not sufficient to explain the one-to-two orders of magnitude difference in the threshold field, as compared to other works utilizing large field enhancement by microstructures. We agree with this point and noticed that one important aspect was not clearly emphasized in the original manuscript. Besides the large total number of electrons due to the large emitting surface and a small average field enhancement due to surface roughness, it was the high sensitivity and efficiency of the detection by the electron multiplier that enabled observing a low threshold field for the electron emission. **In the revised manuscript, the importance of the high detection sensitivity and efficiency are emphasized in the first paragraph of the “Results” section and in the last paragraph of the “Discussion” section.**

2.B

In connection to the previous Fig. 3b (in the revised version: Fig. S4b of the Supplemental document), which shows the polarisation angle dependence of the emission signal in comparison with calculations for different values of the roughness parameter, the Reviewer mentions that the surface roughness does not affect the intensity of electron emission. In order to clarify this point, the characterization of the surface roughness was carried out with significantly improved resolution, as compared to the previously used optical methods. Atomic force microscopy (AFM) was used to measure the surface topography with nm-scale spatial resolution. Subsequently, the AFM data were used to calculate the THz near field and the field enhancement at the surface, the surface density of the emission current, and the effective roughness parameter. The result of this detailed analysis revealed that the effect of surface roughness is not sufficient to explain the measured polarization angle dependence (double maxima) of the emission signal and other effects may contribute. On the other hand, the same analysis also showed that the field enhancement, caused by the surface roughness, strongly influences the electron emission: “hot spots” for electron emission emerge, defining the spatial distribution of the current density, and the scaling with field strength is significantly changed. Thus, in the present case, the surface roughness sensitively affected the intensity of electron emission through local field enhancement, but it was not sufficient to explain the polarization angle dependence.

Related to this new insight, the following changes were made in the revised manuscript. A new paragraph was added to the end of the subsection “THz-waveform-driven surface electron source” in the “Results” section, which introduces the AFM measurements and the near-field calculations. Furthermore, the new Fig. 3 and a related new paragraph were added to the “Discussion” section, which give the details.

Because the revised manuscript includes a much more reliable quantification of the surface roughness based on the AFM topography measurements, **the polarization angle dependent electron emission measurement and its analysis was moved from the main article to the Supplemental document. With this change, the revised manuscript is more clearly focused on the main scientific novelty, the demonstration of the confinement of single-cycle THz-waveform-driven electron emission to one of the two half cycles and the control of the active half cycle by changing the field polarity.**

With these changes, the previous Fig. 3, related to the polarization angle dependence of the electron emission, was moved from the main article to Fig. S4 in the Supplemental document (section “Dependence of electron emission on the THz-field polarisation”). The discussion of this result was rewritten. A quantitative analysis of the surface roughness parameter, based on the AFM data, was included into the Supplemental document (section “Characterisation of the BeO surface”). We note

that the curves in Fig. S4, calculated for different surface roughness values, were normalized to the same maximum for a better comparison to each other and to the experimental data. For more clarity, the normalization is explicitly mentioned in the revised caption of Fig. S4b. Furthermore, one paragraph was added to the section “Theoretical model of the field-driven electron emission” in the Supplemental document on how surface roughness affects the intensity of electron emission.

2.C

According to the Reviewer’s opinion, the electron emission signals from reference substrate Cu and BeO are necessary. Because this point is related to the material composition and structure of the cathode, a detailed inspection of the cathode surface was carried out. This inspection included atomic force microscopy, and focused ion beam milling with subsequent scanning electron microscopy. Based on the inspection results, complemented by technical information published by the dynode manufacturer, we concluded that the observed electron emission signal originated from a BeO surface, which was coated onto a conducting Cu substrate. Thus, the manuscript shows the electron emission signals from BeO (see also our response to the Reviewer’s 3rd point). The manuscript was changed to explain that the electron emission signal originated from a BeO surface coated onto a conducting substrate. The changes include: (i) adding a related sentence to the first paragraph of the “Results” section; (ii) changes in Fig. 1 and its caption; (iii) changing the referencing to the cathode from “Cu-BeO” to “BeO” throughout the main article and the Supplemental document. Finally, we note that studying electron emission signals from other materials, such as pure Cu, is of great interest for possible future applications. However, the comparison of various cathode materials, eventually with different surface roughness conditions, is beyond the scope of the present study. Here, the focus was on demonstrating subcycle surface electron emission and the control of the active half-cycle. In order to keep the scope of the present work focused, we did not extend this study to other materials.

Reviewer point:

3. The thickness of Cu on Cu-BeO is useful. It is important to determine the electron emission mechanism.

Author response:

In order to clarify the coating thickness of the cathode, a trench was cut into its surface by focused ion beam milling and, subsequently, the trench edge was inspected by scanning electron microscopy. The secondary electron images confirmed that the surface has a coating layer and indicate that the coating is about 140 nm thick, with local variations between 100 nm to 200 nm. This information is complemented by the technical documentation from the manufacturer. The cathode in the experiment was the first dynode of a commercial electron multiplier. The material of the dynode was specified by the manufacturer as Cu-BeO, where BeO was coated onto a conducting Cu substrate electrode. Thus, the coating was BeO, rather than Cu, and the THz-field-driven electron emission originated from the BeO coating.

In the revised manuscript, in order to clarify the electron emission mechanism, the cathode composition was explained and referencing to the cathode material was changed to BeO, rather than Cu-BeO. Furthermore, in the newly introduced Fig. 1b, the cathode coating and substrate are clearly distinguished. A new first paragraph was added to the section “Characterisation of the BeO surface” in the Supplemental document, which also cites the relevant technical documentation.

Reviewer point:

4. The experimental data (Figure 3a) is asymmetric about $\phi=0^\circ$, which is attributed to asymmetric condition on surface Cu-BeO. The supplemental data about the dependence of electron emission on azimuthal angle of Cu-BeO should be given, which are important experimental evidence.

Author response:

In the experiment, the THz polarization angle φ was varied. Other angles in the setup geometry (angle of incidence of the THz beam θ , azimuthal angle of the BeO cathode surface) were kept constant, due to constraints in the setup. For example, varying the azimuthal angle would have required rotating the cathode surface about its mean surface normal, rather than about the incoming beam axis, which was not possible. For this reason, the dependence of electron emission on the azimuthal angle could not be measured. However, we have carried out a more accurate characterization of the BeO surface by using atomic force microscopy (AFM, see also our response to Reviewer point 2 above). From the AFM data, the polar and azimuthal angular distributions of the local surface normals were calculated. The distribution of azimuthal angles revealed a certain level of directional anisotropy in the surface topology. This could be a reason for asymmetric conditions on the BeO surface, and have an effect on the polarization angle dependence of the electron emission. However, the in-depth study of surface anisotropy effects is beyond the scope of the present work.

In the revised manuscript, the new Figure S3 in the section “Characterisation of the BeO surface” of the Supplemental document shows the polar and azimuthal angle distributions for an AFM-sampled part of the surface. The anisotropy of the surface is mentioned in the accompanying text, together with possibly being a reason for asymmetric conditions on the BeO surface.

As described above, in connection to the 2nd Reviewer point (Author response 2.B), the presentation and discussion of the polarisation angle dependence of the electron emission was moved to the Supplemental document. With this change, the manuscript became more focused on the main scientific novelty, the demonstration of the confinement of single-cycle THz-waveform-driven electron emission to one of the two half cycles and the control of the active half cycle by changing the field polarity.

Response to Reviewer 2

We thank the Reviewer for the critical reading of the manuscript and for the valuable comments. Furthermore, we appreciate the Reviewer's opinion of finding the manuscript well written and interesting, and the experiments carefully carried out. In our response below, the Reviewer points are typeset in blue and our responses in black. The description of the corresponding changes in the manuscript are typeset in red. We believe that the responses appropriately address the Reviewer's concerns and the revised manuscript can be published in Nature Communications.

Reviewer point:

In the manuscript "Subcycle surface electron emission by strong-field terahertz waveforms" by Shaoxian Li et al., the authors reveal the electron emission from a Cu-BeO surface driven by intense THz pulses. This process was observed at remarkably low field strengths of 40 kV/cm without the need for field enhancements generated by nanostructures, for example. A potential explanation for this low threshold lies in the large number of detected electrons owing to the large area that was illuminated. The single-cycle nature of the THz pulses and the significant change in electron yield upon reversal of the field polarity confirm that the emission of the electrons is confined to the halfcycle exerting a force onto the electrons that points away from the surface. Discrepancies between modelling and experiments suggest that the following halfcycle of the transient may drive some of the electrons back to the surface, thereby lowering the electron yield. The data quality is convincing and sufficient details are given for the reader to be able to reproduce the results, in principle. Even though the manuscript is well written and interesting, and the experiments were carefully carried out, I cannot support publication in Nature Communications for the following reasons:

1. The conclusions drawn from the experimental data remain very vague. To describe the dependence of the detected number of electrons as a function of the electric field, the authors need to make several assumptions. On the one hand, there is a discrepancy (0.43 instead of the expected scaling factor of 0.65) used for the relative field strength of the two half cycles. The authors suggest that this could originate from electrons being driven back to the surface. This hypothesis is, however, not quantified by simulations (based on the time-dependent Schrödinger equation, for example). On the other hand, all electric fields additionally need to be scaled by a factor of two to retrieve the correct work function in the Fowler-Nordheim plot. The authors suggest that the surface roughness could yield the required field enhancement.

Author response:

The first part of the Reviewer point is related to the discrepancy (0.43 instead of the expected scaling factor of 0.65) used for the relative field strength of the two half cycles and the effect of electron backscattering.

In the revised manuscript, a quantitative estimation of the effect of backscattering has been included into the analysis, with an estimation of the backscattering coefficient based on the experimental data, classical trajectory calculations, and the numerical solution of the time-dependent Schrödinger equation. By comparing the measured electron emission signals for the opposite THz field polarities, and taking into account a scaling factor of 0.64 for the relative field strengths of the two half cycles (as expected in accordance with the measured waveform), we found that a backscattering

probability of 9% is consistent with the observations. The scaling factor of 0.43 has been removed from the revised manuscript. Fig. 2b and the related discussion (3rd paragraph in the “Discussion” section) were changed to include both the effect of the measured scaling factor (0.64) for the relative field strengths of the two half cycles, as well as backscattering (rather than introducing a new scaling factor deviating from the measured waveform). Furthermore, a new subsection was added to the section “Theoretical model of the field-driven electron emission” in the Supplemental document, which discusses the calculations based on the time-dependent Schrödinger equation. It also includes two new literature references for comparison with relevant works, which give numerical values for electron backscattering coefficients.

The second part of the Reviewer point is related to the surface roughness and the field enhancement caused by it. This is closely related to the 2nd point of the Reviewer and we give a more detailed response there. In brief, based on atomic force microscopy (AFM) of the cathode surface, it is pointed out that the newly added surface characterization measurements and the subsequent numerical analysis confirmed that the surface roughness can yield the required field enhancement.

In the revised manuscript, a new paragraph was added to the end of the subsection “THz-waveform-driven surface electron source” in the “Results” section, which briefly describes the characterization of the cathode surface topography and the subsequent numerical simulations to calculate the near-field enhancement. A new paragraph with the new Fig. 3 were added to the “Discussion” section, which discuss in more detail the characterization of the local field enhancement and how it influences the electron emission. The new subsection “Calculation of the THz near field” was added to the “Methods” section. Finally, the section “Characterization of the BeO surface” in the Supplemental document has been rewritten and significantly extended.

Reviewer point:

2. The field enhancement resulting from the surface roughness is not quantified reliably on the relevant length scales. The maxima in the polarization dependence on the number of emitted electrons suggest that a significant surface roughness is present. To describe their data, a characteristic parameter of $\sim 15^\circ$ is required. Yet, the authors only provide optical images on the micrometer-scale resulting in a significantly smaller value of 2.6° , for which only a single maximum is expected according to their theoretical model. For a high-ranking journal such as Nature Communications, topography measurements on the nanoscale (atomic force microscopy etc.) should be conducted for a reliable quantification of the roughness parameter. Based on these insights, numerical simulations (finite element method, for example) should subsequently be performed to quantify the resulting field enhancement in the THz spectral range. This would also resolve some of the points addressed in (1).

Author response:

The Reviewer point refers to two related issues: on the one hand, to the reliable quantification of the field enhancement on the relevant length scales, and, on the other hand, to the reliable quantification of the roughness parameter.

In order to reliably quantify the surface roughness on the relevant length scales, topography measurements were conducted using atomic force microscopy (AFM). Based on the surface topography data, subsequent numerical simulations (based on the finite element method) were performed to quantify the resulting field enhancement in the THz range. It was found that the field

enhancement strongly varies on the sub- μm scale. Local maxima of the field enhancement factor up to 2 to 4 were found at granular structures, with stronger surface curvatures. Importantly, this is in very good agreement with our previous estimation of about 2 for the effective field enhancement factor, which was solely based on fitting the experimental data with a Fowler-Nordheim model. (The effective field enhancement is somewhat smaller than the highest local maxima, because it is an average value over the part of the surface where significant electron emission takes place.) This also resolves some of the points addressed in the 1st Reviewer point. In conclusion, the newly added surface characterization measurements and the subsequent numerical analysis clearly confirmed that the surface roughness could yield the required field enhancement.

The related changes in the manuscript were described in our response to the 1st Reviewer point. For completeness, these are repeated here. **In the revised manuscript, a new paragraph was added to the end of the subsection “THz-waveform-driven surface electron source” in the “Results” section, which briefly describes the characterization of the cathode surface topography and the subsequent numerical simulations to calculate the near-field enhancement. A new paragraph with the new Fig. 3 were added to the “Discussion” section, which discuss in more detail the characterization of the local field enhancement and how it influences the electron emission. The new subsection “Calculation of the THz near field” was added to the “Methods” section. Finally, the section “Characterization of the BeO surface” in the Supplemental document has been rewritten and significantly extended.**

Furthermore, the topography data from the AFM measurements were also used to calculate the distribution of the polar and azimuthal angles of the local surface normal, and to quantify the surface roughness parameter. The polar angle distribution is dominated by small ($\lesssim 5^\circ$) angles, but large (10° - 20°) angles also occur. Locations of larger polar angles correlate with locations of field enhancement maxima (because of stronger surface curvatures at such locations), and consequently, with the electron-emitting “hot spots” (see Fig. 3 and its discussion in the main text). For this reason, an effective roughness parameter was introduced and its value evaluated, whereby the averaging was confined to the emitting “hot spots”. **The result of this study and the corresponding discussion was included into the Supplemental document (section “Characterisation of the BeO surface”).**

As one conclusion of the investigations described above, it was found that the polarization angle dependence of the electron emission signal is not well suited to reliably quantify the surface roughness (through the surface roughness parameter $\bar{\alpha}$, defined in the Supplemental document). Because the revised manuscript includes a much more reliable quantification of the surface roughness based on the AFM topography measurements, **the polarization angle dependent electron emission measurement and its analysis was moved from the main article to the Supplemental document. With this change, the revised manuscript is more clearly focused on the main scientific novelty, the demonstration of the confinement of single-cycle THz-waveform-driven electron emission to one of the two half cycles and the control of the active half cycle by changing the field polarity.**

Reviewer point:

- 3. How accurately is the work function of Cu-BeO known? The authors only refer to a personal communication with the manufacturer Hamamatsu? Can the possibility of a significantly smaller work function (<4.1 eV) reducing the required field enhancement be ruled out? Additional AFM measurements and numerical simulations (see my previous point) would also help clarify this point. Are there any measurements or literature values?**

Author response:

The material of the cathode was BeO, coated onto a Cu substrate (see “Methods: Interaction chamber” in the revised manuscript and the Supplemental document). According to our search, data on the work function of BeO in the scientific literature and other publicly accessible resources are sparse. In the revised manuscript, besides the mentioned personal communication with the manufacturer Hamamatsu ($W = 4.1$ eV), **a new reference to a compilation of measured work function data was added to the revised manuscript, which includes BeO and is available online.** This latter resource gives the range of 3.80 – 4.70 eV, which includes the value given above. In this context we also note that technical application notes by Hamamatsu mention that electron multiplier dynodes use materials with large work function to reduce the dark current. Furthermore, the additional AFM measurements and numerical simulations (see the previous point) clarified the value of the field enhancement. A work function of about 4 eV is consistent with an effective field enhancement of about 2, as it was pointed out in the manuscript.

Based on this, we concluded that a work function significantly smaller than about 4 eV is not consistent with the observations and calculations (electron emission data, field enhancement) and the other available information (work function data, technical information from the manufacturer, personal communications), and therefore it was ruled out.

Reviewer point:**4. Some minor points:**

- The electric field in Fig. 1b could be calibrated to kV/cm
- A justification why Cu-BeO was used is missing
- In the introduction (line 30), references should be added for the extensive studies that are mentioned

Author response:

- **In the revised manuscript, the THz waveform is shown in Fig. 1c, because a new Fig. 1b was added. The electric field was calibrated to the maximum value used in the experiment and is given in kV/cm. Furthermore, the field maxima in the leading and trailing half cycles are explicitly shown and more clearly defined. For clarity, only the waveform corresponding to the cathode position is shown.**
- The observed electron emission signal originated from a BeO surface, coated onto a Cu substrate (see the response to the 3rd Reviewer point above). Such and similar surfaces are frequently used in devices based on electron emission, such as electron multipliers, and therefore it has significant technological relevance. Importantly, this choice of the cathode surface enabled a simple experimental configuration where the cathode was optimally integrated with the detector for a highly efficient and sensitive detection (electron multiplier) of the released electrons. **A note on the reason of choice was added to the first paragraph of the subsection “THz-waveform-driven surface electron source” in the revised manuscript. In connection to this, the importance of the high detection sensitivity was explicitly emphasized in the last paragraph of the “Discussion” section.**
- **References to the mentioned previous studies were added to the first sentence of the “Introduction” section. Given the huge number of relevant works published over decades and the length limitation of this work, references to only a few examples could be inserted.**

Reviewer point:

While the manuscript provides new and interesting results, the open questions and unknowns listed above are limiting the potential impact of this work. Unless the authors can significantly revise the manuscript and provide a more quantitative explanation of their experimental results, I cannot support publication in Nature Communications, unfortunately.

Author response:

We appreciate the Reviewer's opinion of finding the results new and interesting. We believe that the open questions and unknowns, which were limiting the potential impact of this work, could be eliminated by the major revision. With the help of the newly added measurements and simulations, a more quantitative explanation of the experimental results could be given. We thank the Reviewer for the valuable and constructive criticism, which helped us to improve significantly the manuscript.

Reviewers' Comments:

Reviewer #1:

The manuscript entitled "Subcycle surface electron emission driven by strong-field terahertz waveforms" by S. Li et al. reported interesting and significant experiments. In addition, the authors demonstrated the confinement of single-cycle THz-waveform-driven electron emission to one of the two half cycles and the control of the active half cycle by changing the field polarity. The large total number electron emission from BeO surface at threshold field of about 40 kV/cm induced by effective field enhancement due to surface roughness. In conclusion, The experimental results worth to be published in Nature Communications.

Reviewer #2:

Remarks to the Author:

The authors have revised the manuscript, substantially improving it in various aspects:

- The authors have performed additional AFM measurements to extract the surface roughness as suggested. Using the results as input for their numerical simulations, the manuscript now includes maps of the local field enhancement. The factor of 2 in field enhancement - previously simply assumed - is now well justified.

- In the initial version of the manuscript, an additional scaling factor for the two half cycles was used. By performing TDSE simulations, the authors extract a backscattering rate of $\sigma=9\%$ for the weaker half-cycle. Rescaling the number of electrons by this correction factor, the yield for the weaker half-cycle now also matches the expectations for strong fields.

- The interesting polarization-dependence of the electron emission has been moved to the SI as it still cannot be reconciled with predictions based on straightforward models.

- Furthermore: It is now clear why BeO was used in the study; Additional details concerning the work function of the material are provided; and the electric field is now given in absolute units in every figure panel.

I can now support publication in Nature Communications after a few minor points have been addressed:

- Fig. 3: A label for the colorbar in panel d is missing. How was this data obtained? Was the data in panel b plugged into the Fowler-Nordheim formula to derive a current from a local electric field? I suggest to provide a few more details on the simulations as the caption and Methods section are rather short.

- Fig. 2b: The slope for the weak half-cycle (empty blue squares) still does not match the expectations. Can the authors provide possible explanations?

Response to Reviewer 2

We thank the Reviewer once again for the critical reading of the manuscript and for the valuable comments. Furthermore, we thank for the Reviewer's support of publication of the revised manuscript in Nature Communications. We believe that the responses given below and the related modifications in the manuscript appropriately address the Reviewer's minor points. In our response, the Reviewer points are typeset in blue and our responses in black. The description of the corresponding changes in the manuscript are typeset in red.

Reviewer point:

The authors have revised the manuscript, substantially improving it in various aspects:

- The authors have performed additional AFM measurements to extract the surface roughness as suggested. Using the results as input for their numerical simulations, the manuscript now includes maps of the local field enhancement. The factor of 2 in field enhancement – previously simply assumed – is now well justified.
- In the initial version of the manuscript, an additional scaling factor for the two half cycles was used. By performing TDSE simulations, the authors extract a backscattering rate of $\sigma=9\%$ for the weaker half-cycle. Rescaling the number of electrons by this correction factor, the yield for the weaker half-cycle now also matches the expectations for strong fields.
- The interesting polarization-dependence of the electron emission has been moved to the SI as it still cannot be reconciled with predictions based on straightforward models.
- Furthermore: It is now clear why BeO was used in the study; Additional details concerning the work function of the material are provided; and the electric field is now given in absolute units in every figure panel.

I can now support publication in Nature Communications after a few minor points have been addressed:

- Fig. 3: A label for the colorbar in panel d is missing. How was this data obtained? Was the data in panel b plugged into the Fowler-Nordheim formula to derive a current from a local electric field? I suggest to provide a few more details on the simulations as the caption and Methods section are rather short.

Author response:

Fig. 3 was reformatted. Now, all panels include x and y scales and labels, as well as greyscale (panel a) or color (panels c-d) bars with labels.

The local current density data in Fig. 3d were calculated with the Fowler-Nordheim formula, from the calculated local electric field data. The local electric field was obtained by using the field enhancement according to Fig. 3b. A polarity angle of $\varphi = 0^\circ$ (trailing half cycle active) and an electric field amplitude of $E_2 = E_0 = 105$ kV/cm were assumed. Clarifying details on the current density simulations were added to the rewritten paragraph in the main text related to Fig. 3, and to the caption of the figure. One related sentence was added to the section Methods: Calculation of the THz near field.

Reviewer point:

- Fig. 2b: The slope for the weak half-cycle (empty blue squares) still does not match the expectations. Can the authors provide possible explanations?

Author response:

We thank the Reviewer for this hint. Even though we are not quite sure what is meant by the term "expectations" in this context, we agree that the exact slope values should be investigated in more detail. However, such an interesting phenomenon deserves more attention, which would require additional measurements with highly advanced instrumentation such as, for example, a NanoESCA instrument with full k-space resolution of field-emitted electrons. Without such tools, we can not draw relevant experimental conclusions on slope differences. Because of the complexity of such an experiment, this should be the subject of a future project. We would like to stress, however, that this particular observation does not affect any of the conclusions of our paper. The most conspicuous feature, that field emission is so much suppressed by the polarity flip of the THz pulse, is upheld and this is a feature that has never been observed in optical-domain measurements before.

Various factors can influence the slope of the field dependence of the electron emission signal in the Fowler-Nordheim plot. On a hypothetical basis, we can think about the angle of incidence of the THz beam, the field dependence of the backscattering probability for $\varphi = 180^\circ$ polarity (the returning electron energy depends on the field strength), etc. as a reason for the observed difference. However, we decided not to formulate these hypotheses in the revised manuscript. In addition, **we would like to note that since the calculated line with $W = 2.5$ eV might be misleading in the interpretation of Fig. 2b, we removed this from the revised version and the 5th paragraph in the "Discussion" section was shortened and rewritten.**

Reviewers' Comments:

Reviewer #2:

Remarks to the Author:

I have no additional comments and fully support publication of the manuscript now.